# Redbay (*Persea borbonia* L. Spreng) Seedling and Sapling Growth and Recruitment Near Trees with and without Putative Resistance to Laurel Wilt Disease

Matthew Eicholtz [1], Jason Smith [2] and Jason Vogel [1,*]

1   School of Forest, Fisheries and Geomatics Sciences, University of Florida, Gainesville, FL 32611, USA; meicholtz914@gmail.com
2   Department of Biology, University of Mount Union, Alliance, OH 44601, USA; smithjas@mountunion.edu
*   Correspondence: jvogel@ufl.edu; Tel.: +1-352-846-0890

**Abstract:** Laurel wilt, a fungal disease (*Harringtonia lauricola* T.C. Harr., Fraedrich and Aghayeva) spread by the Asian redbay ambrosia beetle [*Xyleborus glabratus* Eichhoff (Coleoptera: Curculionidae: Scolytinae)], presents an imminent threat to North American members of the Lauraceae family, having caused extensive mortality in several species, especially redbay (*Persea borbonia* L. Spreng). Varying levels of disease resistance have been recorded in redbay under controlled conditions. To investigate if previously monitored putatively resistant field redbays have influenced the regeneration and survival of conspecifics within the surrounding 0.08 ha, a survey was conducted in 2018-19 and compared to similar data collected years prior (2008-09, 2013) along the coasts of Florida, Georgia, and South Carolina, United States. Plots were originally established at six disease-infested study sites around large redbay (>7.5-cm diameter at breast height (1.37 m) (DBH)) that had survived the initial laurel wilt disease epidemic that began in approximately 2007. In 2018-19, a subset of 61 plots within 16 m of the original "survivor" redbay were recorded and compared to previous surveys. Among the original redbay selected for resistance, 22 of 61 (36%) survivors across all sites were alive in 2018 with survival rates varying from 0 to 70% between survey periods (average mortality 3.6%/year). Trees that died in years since 2008-09 had their plots reclassified as susceptible or "suscepts". Changes in mean quadratic diameter at 1.37 m of redbay plots near survivors were significantly greater than those near suscepts, and in 2018-19, the average diameter of redbay near survivors was 7.62 cm vs. 4.90 cm for suscepts. The diameter distribution of dead and live redbay in the whole population showed a decrease in live individuals surviving past 8 cm DBH in 2018–2019, but 20 of 22 survivor candidate trees were larger than 8 cm DBH. Regeneration was occurring both clonally and sexually and tended to be greater near suscepts, but midstory resprouts per hectare and understory seedlings interacted significantly with the site and the latter differed between sites. These findings indicate that redbay is regenerating in these ecosystems, and disease resistance may allow for increased average tree size for some individuals, but an upper size threshold of around 8 cm DBH may still exist for much of the population. In addition, the importance of site variables in regeneration was apparent, making either local genetic or environmental effects an important topic for future research. Continuing to monitor these survivors while locating new candidates for disease screenings and breeding, preventing the introductions of new strains of *H. lauricola*, shedding light on the nature of resistance and its heritability, and initiating outplanting trials with resistant germplasm are instrumental steps in bringing redbay back to prominence in its historical range.

**Keywords:** laurel wilt; disease resistance; reforestation; regeneration; redbay; Lauraceae; forest health; management

## 1. Introduction

For over a century, anthropogenic activities have facilitated the biological invasions of exotic pests and pathogens into new habitats throughout the world [1–6]. In North

America, introduced pathogens have caused significant population declines in numerous tree species [2,5–10]. Laurel wilt is a vascular wilt disease that has plagued various North American members of the Lauraceae dating back to the 2002 introduction of the Asian red-bay ambrosia beetle [*Xyleborus glabratus* Eichhoff (Coleoptera: Curculionidae: Scolytinae)] into the southeastern United States [11–15]. *X. glabratus* serves as the primary vector of the disease by spreading its fungal symbiont, *Harringtonia lauricola* T.C. Harr., Fraedrich and Aghayeva, to laurel hosts upon initial attempts to colonize them and subsequently eliciting defensive responses in the host, ultimately leading to mortality [11–13,16–18].

Redbay (*Persea borbonia* L. Spreng.), an aromatic, evergreen member of the Lauraceae native to the southeastern Atlantic and Gulf Coastal Plains has experienced unprecedented mortality due to laurel wilt (Figure 1) [11,19,20]. A once abundant and characteristic member of coastal maritime forest canopies and subcanopies, redbay is a culturally and ecologically significant species within its range, with notable usages medicinally and traditionally in certain indigenous tribes, and it is also a component in regional cuisines and a wood source for constructing cabinets and boats [21]. Its tolerance for a variety of light and soil conditions and evergreen foliage has made it a popular landscape tree, and several species of wildlife utilize redbay as a food and habitat source, with it being the preferred larval host of the Palamades swallowtail [*Papilo palamades* (Drury, 1773)] in particular [21–23].

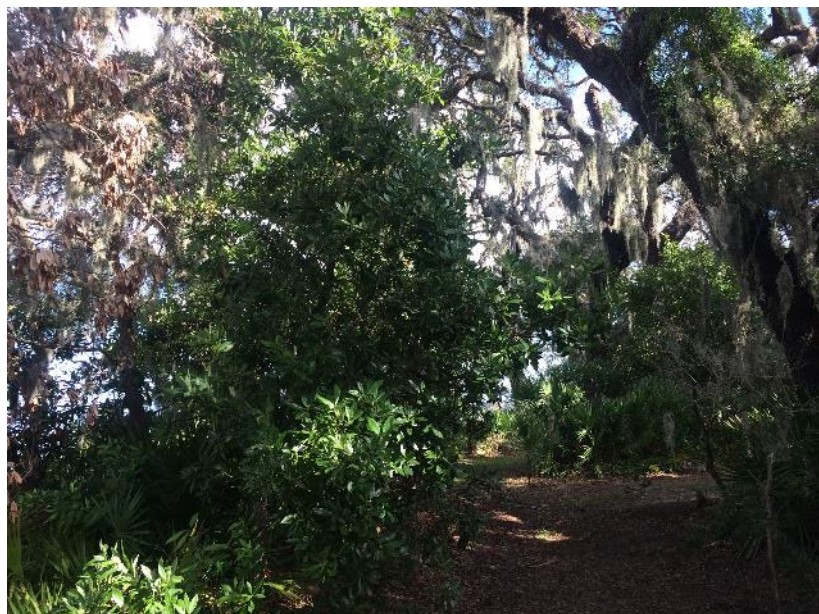

**Figure 1.** Healthy redbay at Cumberland Island National Seashore. A redbay killed by laurel wilt appears to the left of it. Photo courtesy of Matthew Eicholtz.

At some sites within redbay's historic range, over 90% mortality has been reported in stems > 2.5 cm diameter at breast height (DBH) within several years after initial signs of laurel wilt [11,19]. In 2017, it was reported that roughly 300 million redbay had been killed by the disease [20]. Given its once prominent presence throughout its range, the rapid elimination of mature redbay stems in coastal forests has significantly altered community composition, resulting in ecological processes being influenced by these changes [19,20,23–26].

Many of the existing management strategies are either preventative or focused on limiting the spread of laurel wilt once symptoms have been observed or if disease vectors have been detected [27–30]. While certain fungicidal methods are effective, they may be limited to use on high-value individual trees as they are neither feasible nor cost-effective in natural forest settings [28,29]. Furthermore, no management strategies exist for saving trees that have already been infected Discovering disease resistance in redbay populations may be an effective and practical method for reforestation efforts and long-term survival of

the species [28,31]. As in the cases of species, such as American chestnut [*Castanea dentata* (Marsh.) Borkh.], American elm (*Ulmus americana* L.), Port-Orford-cedar [*Chamaecyparis lawsoniana* (A. Murray bis) Parl.], and ʻōhiʻa (*Metrosideros polymorpha* Gaudich.), the discovery of disease resistance can provide an opportunity to restore species that are unable to naturally regenerate effectively due to overwhelming pathogenic pressure [8,32–34]. While the deployment of resistant hosts has been suggested as a plausible management strategy [28], studies on the long-term survival of resistant redbays in situ are non-existent.

Between 2007 and 2009, a survey (survey one) for resistant redbay at six laurel wilt-infested sites along the Atlantic Coastal Plain yielded the identification of 84 mature, healthy individuals (>7.5 cm DBH), which would serve as candidates for future monitoring and disease screening [31,35]. Demographic surveys for redbay saplings (>2.5 cm DBH) were conducted within the surrounding 0.08 ha of each of the candidates shortly after their identification [31,35]. Cuttings from 38 of these individuals were vegetatively propagated and screened to determine their respective responses to *H. lauricola* inoculation (Figure 2) [31,35]. While many genotypes succumbed to artificial inoculations, several showed varying levels of tolerance to the pathogen, with some surviving multiple, would-be lethal doses of inoculum. Unfortunately, there was also genetic variation in how well individual trees could be clonally propagated, leaving it unknown as to whether *H. lauricola* resistance was occurring in situ. These findings warranted further investigations into the fate of the candidates in situ as well as whether the presence of a resistant sapling affected adjacent sapling survival beyond the accepted size threshold of susceptibility to *X. glabratus* attack. Moreover, if these still living candidates ("survivors") have significantly different plot demography than those that have died ("suscepts"), this could provide preliminary evidence of heritable resistance and subsequently encourage investigations into genetic linkages among trees and seedlings.

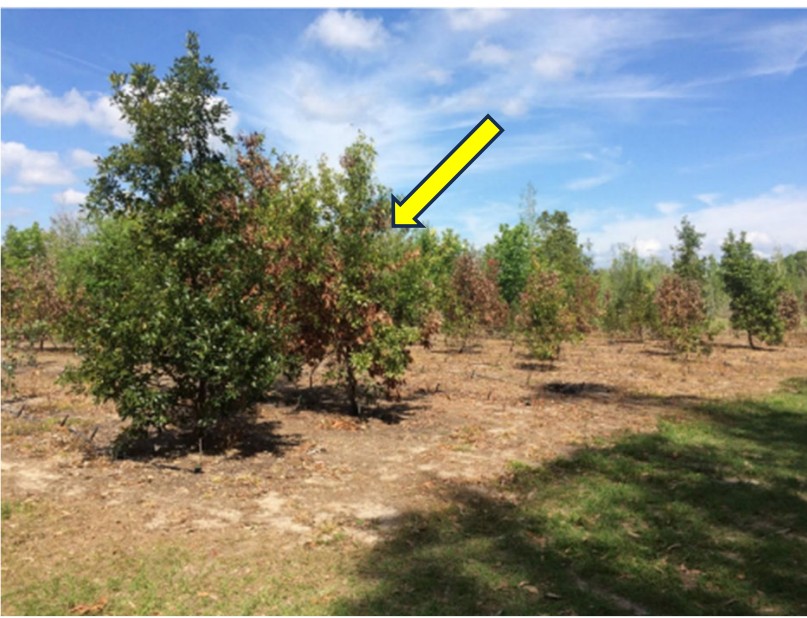

**Figure 2.** Redbay clones inoculated with *Harringtonia lauricola*. These clones were propagated using cuttings taken from candidates at study sites and showed different levels of resistance during laurel wilt disease inoculation tests, as denoted by brown foliage (yellow arrow). Photo courtesy of Kathy Smith, University of Florida Plant Science and Research Education Unit in Citra, FL.

Between 2018 and 2019, a follow-up observational study (survey two) was undertaken to record changes in candidate growth and survival as well as plot demography, specifically the size and number of saplings and understory seedlings surrounding each candidate. The objective of this study was to use trends in redbay survival and recruitment over time in areas heavily affected by laurel wilt to investigate if genetic resistance may be contributing

to the regeneration, growth, and survival of the species at these sites. More specifically, by shedding light on these associations, this study will serve to offer context to questions about the overall efficacy of redbay reforestation using germplasm from the survivors. Furthermore, by identifying which individuals continue to survive to the present, this study can help to pinpoint the most disease-tolerant genotypes, thus giving them a greater focus in lab assays and molecular analyses and helping to identify the genes responsible for resistance.

## 2. Materials and Methods

### 2.1. Survey Sites and Initial Data Collection

In advance of survey one, six locations (Figure 3a) along the Florida, Georgia, and South Carolina coasts were selected and surveyed for large (>7.5 cm DBH) live redbay trees to be selected as candidates for propagation and subsequent laurel wilt disease screenings [31,35]. The Florida sites were located at Fort George Island Cultural State Park (FG), near the Kingsley Plantation, and Fort Clinch State Park (FC). The Georgia sites were on St. Catherine's Island (SCI) and the Cumberland Island National Seashore CINS). The South Carolina sites were located at Hunting Island State Park (HI) and Edisto Beach State Park (EB). These locations are dominated by maritime coastal hammock forests, which characteristically feature dense overstory assemblages of live oak (*Quercus virginiana* Mill.), cabbage palm (*Sabal palmetto* (Walt.) Lodd.), and redbay. Fraedrich et al. found that 92.4% of 132 redbay stems > 2.5 cm DBH had been killed by laurel wilt between July 2005 and October 2006 at FG during the initial epidemic [11]. The prolific mortality in these site's once-abundant redbay populations allowed for a practical on-site method with which to select candidates for monitoring and disease screening (Figure 3b).

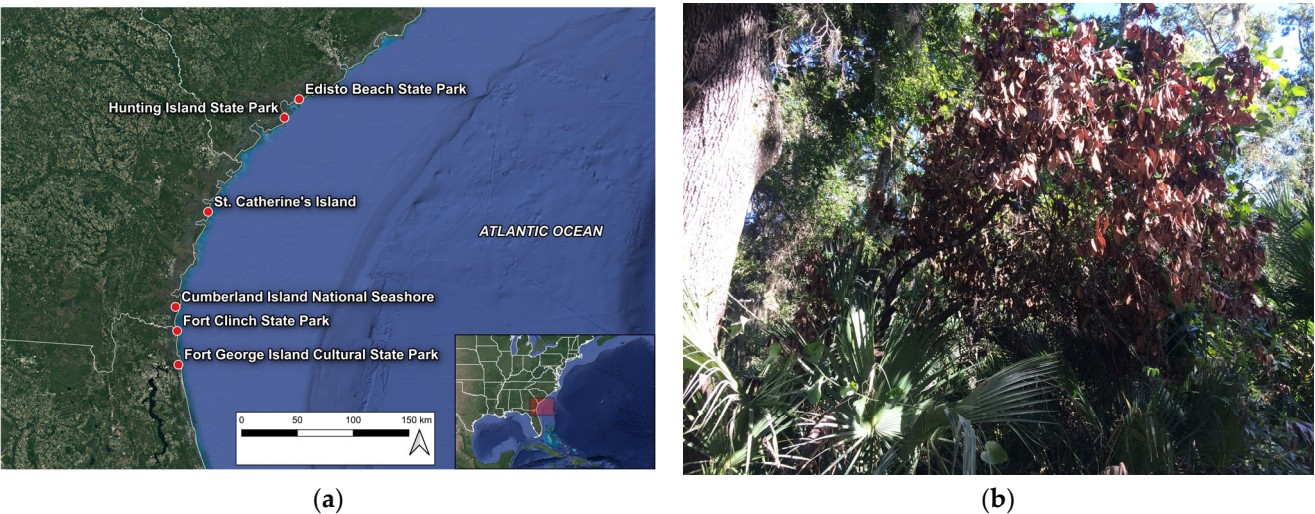

(**a**)                                                          (**b**)

**Figure 3.** (**a**) Map showing the six survey locations along the southeastern Atlantic Coastal Plain, USA. The red-shaded area shows the location of the research area within the region. (**b**) Redbay at Fort George Island Cultural State Park showing characteristic crown wilt after laurel wilt infection. Coordinates for candidate locations are available upon request. Photo courtesy of Matthew Eicholtz.

During survey one, between 10 and 22 candidates at each of the 6 sites (84 in total) were identified, tagged with permanent metal markers, and selected for monitoring (Figure 4d). The locations of these trees were recorded with a Global Positioning System (GPS) GPSmap 60Cx device (Garmin, Olathe, Kansas). Distance between candidates ranged from <32 m to 3.2 km. Overlap was reported in plots at St. Catherine's Island due to the proximity of the candidates during the first survey. At each site, 0.08 ha plots were established surrounding the candidate trees. Within each plot, demographics including candidate DBH, the number and DBH of living and dead saplings, and laurel wilt incidence (% of infected redbay

saplings) were recorded. A 2013, monitoring effort provided updates on changes in percent mortality among the candidates [35].

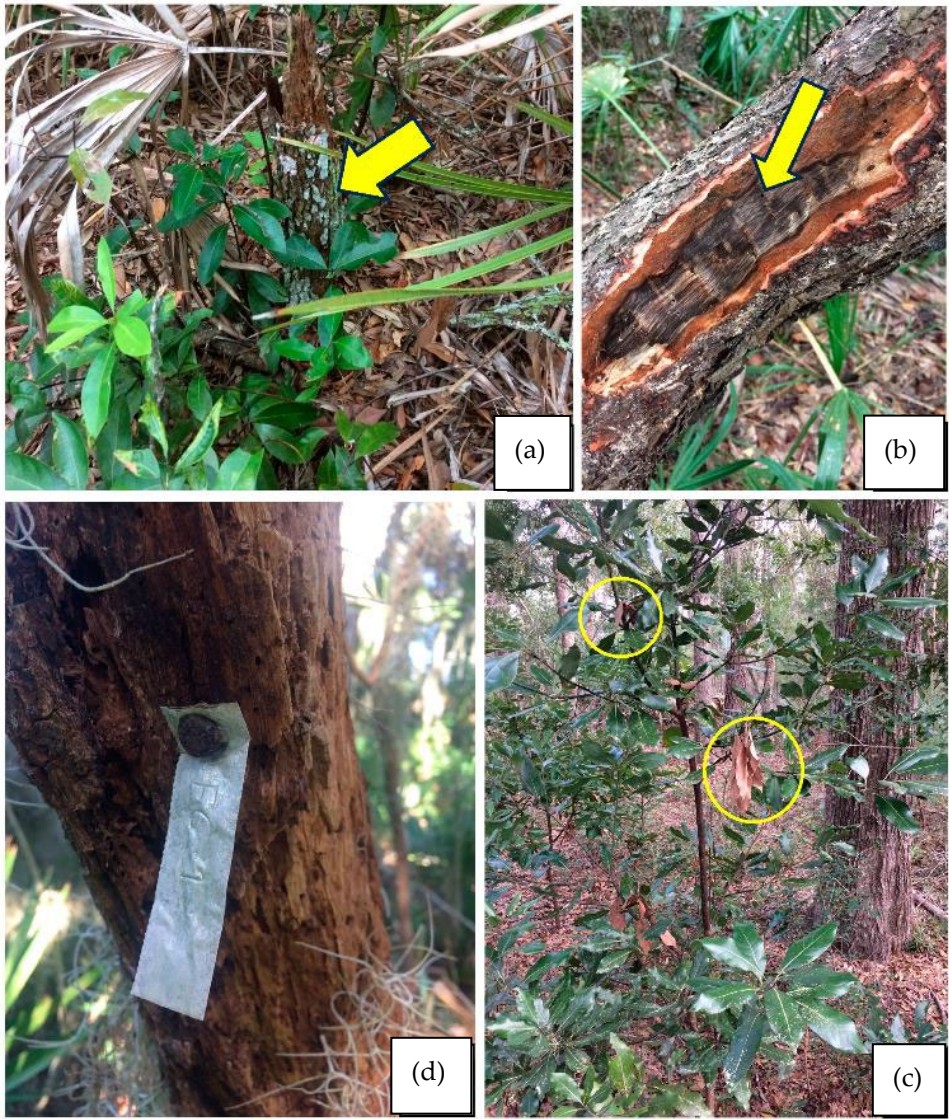

**Figure 4.** (**a**) A proliferation of redbay sprouts emanating from the base of a redbay stump (denoted by arrow). Sprouts can be observed emanating from both live and dead redbay. (**b**) Sapwood staining visible on a dead redbay. Arrow denotes beetle entrance holes. (**c**) Understory redbay showing *X. compactus* damage. Contrary to laurel wilt, *X. compactus* causes localized and sporadic dieback, occurring on redbays at all life stages. (**d**) Dead redbay in later stages of decomposition at Fort Clinch State Park showing permanent metal marker. Photos by Matthew Eicholtz.

### 2.2. Survey Design

The second survey commenced between October 2018 and March 2019, and GPS coordinates were used to locate the candidate plots. Based on their live or dead status, candidates were now either labeled survivors or suscepts, and survivor DBH, sapling DBH, number of saplings, understory seedlings, and understory seedling size class were recorded. The metal tags that had been used to mark the trees during the initial surveys were used to verify the identity of each candidate. In the cases of complete suscept decomposition, the exact location was estimated by utilizing a combination of GPS, sapling spatial distribution data from the 2013 monitoring, and hand-written maps from the time of the initial surveys. At times, suscepts would be found in decomposed states in which sapwood staining

could not be observed (Figure 4d). Monitoring information from 2013 assisted in the confirmation of candidates that were already killed by laurel wilt. Assumptions on the source of some suscepts' mortality were made given the history of laurel wilt at these sites, with the evidence of still-existing disease pressure basal sprouts or stumps denoting the previous presence of a redbay, as well as a lack of other known causes of mortality in the literature [20]. Moreover, not all survivors, while theorized to have some level of genetic resistance, given their continued survival and empirical evidence, have been confirmed as being officially resistant.

Plots were delineated at 4 m intervals radiating from the center tree in the four cardinal directions. In each cardinal direction, 2 m × 16 m transects for understory sampling were established. Once plot boundaries and transects were demarcated, surveys of the plots were carried out.

### 2.2.1. Redbay Sapling Survey

The same sampling protocol used by Hughes [35] was used for surveying redbay saplings. Standing redbay > 1.5 m tall with a DBH > 2.5 cm qualified as saplings. Saplings, both live and dead, within each plot were identified and assessed for laurel wilt infection based on the presence of observable crown wilting or streaked sapwood [Figure 4b]. The black twig borer [*Xylosandrus compactus* Eichhoff (Coleoptera: Curculionidae: Scolytinae)] is a non-lethal pest that also affects redbay by causing scattered small branch and twig dieback (Figure 4c) [28]. Observing sapwood staining allowed for a method to distinguish the potential ambiguity caused by *X. compactus* damage. *X. compactus* populations have not been assessed at any of these sites. The DBH and spatial distribution of saplings within each plot were recorded. Basal sprouts found at the base of dead redbay stumps and snags and live saplings were included in the survey if they met the size criteria (Figure 4a). Seedlings and sprouts emanating from either roots or stumps were distinguished. Dead trees were tallied in the survey if they remained standing with dead foliage still present. Forest Inventory Analysis (FIA) standards for measuring DBH were followed for trees with forks. The distance and direction of each of these saplings from the center tree were measured using a G-Force 1300 ARC Laser Rangefinder (Bushnell, Overland Park, Kansas) and a compass, respectively. If dense vegetation prevented the range finder from obtaining an accurate reading, then a tape measure was used to determine the distance from the center tree. The size and number of saplings in each plot were used to calculate quadratic mean diameter at DBH (QMD), basal area (BA), and saplings per hectare (TPH).

### 2.2.2. Redbay Seedling Survey

Surveys of understory seedlings took place within the directional transects that had been established in each plot. Understory seedlings were classified as any redbay < 1.5 m tall or <2.5 cm DBH. A height pole segmented into 30 cm intervals was used to place each seedling into a size class. Beginning at 1 cm height, size classes increased sequentially every 30 cm (e.g., 1–31 cm). Seedlings were distinguished from resprouts based on the morphology of the stem and the origin of the shoot. Sprouts either originated as root/basal shoots from a seedling, the stump of a dead redbay, or the base of a live- or dead-standing tree (Figure 4a). The number of sprouts around both stumps and seedlings was recorded, and the range of the sprouts' size classes was determined. Black twig borer damage on seedlings and resprouts was noted, but this species does not typically kill redbay, so these data are not reported here.

### 2.3. Statistical Analysis

R version 3.6.0 (R Core Team 2021) was used to analyze changes in the number and sizes of live redbay at these sites between each survey year. Analysis of variance (ANOVA) was used to compare temporal changes in the TPH and QMD between plots with and without survivors at these sites. An ANOVA was also used to determine the responses of TPH of plot saplings from different origins (resprouts and seedlings) based on site and

the presence of a survivor. An ANOVA using the presence of a survivor as a predictor for understory seedlings (SDu) at each site was also carried out. A log transformation was used to improve the normality of Sdu. The TPH from each survey year was square root transformed to improve the normality of ΔTPH. Midstory seedling TPH and midstory resprout TPH were square root transformed to improve normality. Pairwise comparisons of interactions were calculated using the "emmeans" package.

## 3. Results

### 3.1. Field Surveys

3.1.1. Redbay Sapling Survey

Since the start of survey one, 22 of the 61 (36%) original survivors remained alive in 2018 across all study locations (Figure 5), but survival varied among locations. At FG, six survivors were alive while at FC, each of the original seven candidates had died. At CINS, seven survivors remained, and at SCI, six survivors remained. At HI, two survivors remained, and at EB, only one of the original survivors was alive in 2018-19.

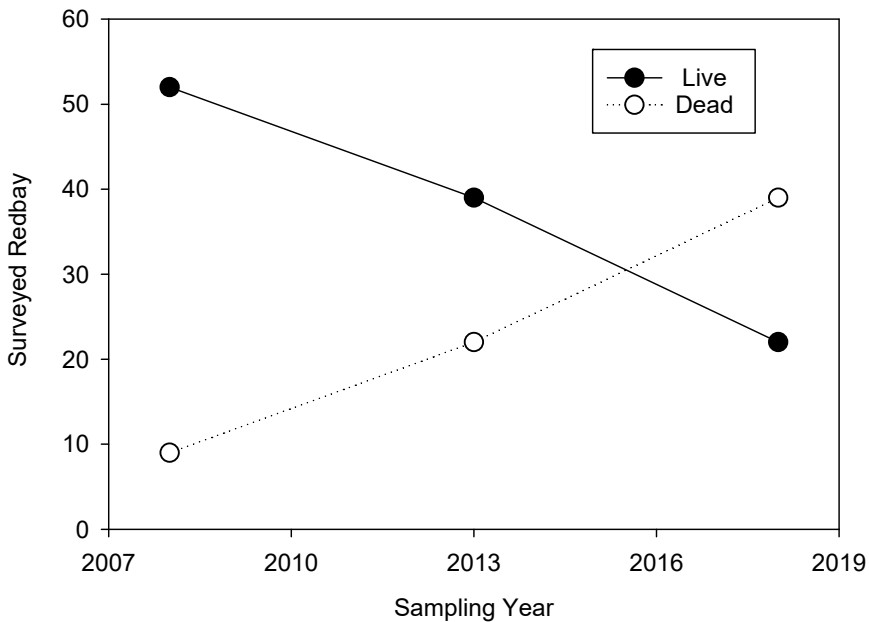

**Figure 5.** Survival over time of the original candidate-resistant trees at five-year intervals.

The DBH of saplings ranged from 8.0 cm to 20.9 cm across sites and years. The remaining survivors' average DBH increased by 2.60 cm across all sites for a growth rate of 0.26 cm/year. The average increase ranged from 1.90 cm (EB) to 4.10 cm (SCI) (Figure 6).

Across sites, the changes in the quadratic mean diameter (ΔQMD) of saplings were found to be significantly higher near survivors compared to suscepts ($p$ = 0.0027) (Table 1). Significant interactions with site and survivors were found for changes in sapling TPH ($p$ = 0.0105) (Table 1), reflecting that one site (HI) showed an opposite trend than the others where survivors had greater sapling TPH around them (Table 2).

**Table 1.** Contrasts of temporal changes (Δ) in redbay saplings per hectare (TPH) and their quadratic mean diameter (QMD) (cm) between two sampling periods for plots around survivor and suscept trees. Plot values indicate means ± SE. Values are significant (*) at the level of $p$ < 0.05.

|  | Survivor Plots | Suscept Plots | Survivor Effect | Site Effect | Survivor × Site Interaction |
|---|---|---|---|---|---|
| ΔTPH | 58.4 ± 21.7 | 60.8 ± 14.1 | $p$ = 0.758 | $p$ = 0.00599 * | $p$ = 0.01050 * |
| ΔQMD | 1.36 ± 0.394 | −0.675 ± 0.448 | $p$ = 0.00269 * | $p$ = 0.95495 | $p$ = 0.87504 |

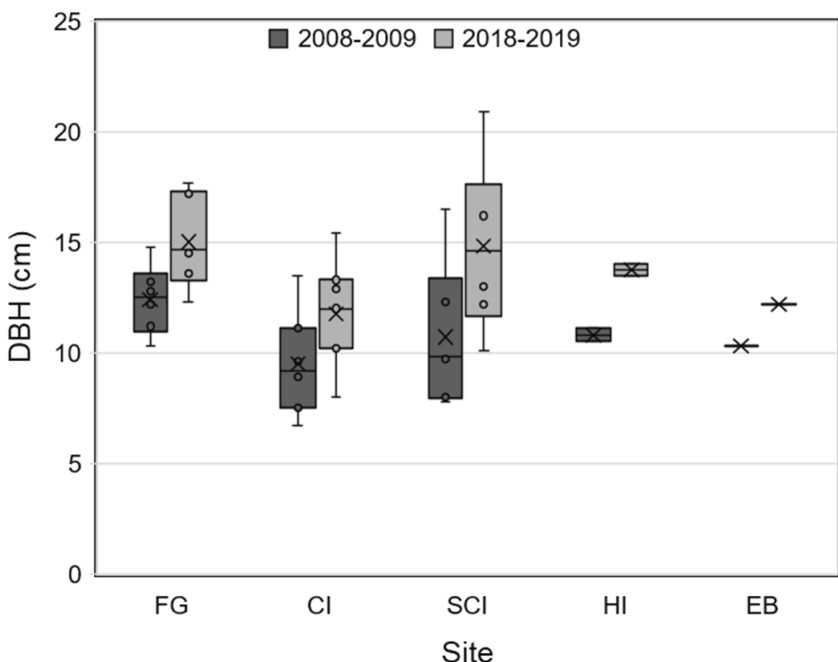

**Figure 6.** Mean DBH of site survivors for the two surveys. Fort Clinch was not included, as all candidates died at the site. EB only had one surviving tree. Mean ± SE.

**Table 2.** Pairwise plot comparisons between survivor and suscept plots for changes in saplings per hectare recorded during the field survey. The FC site had no survivors. Values are significant (*) at the level of $p < 0.05$.

|  | Site | Contrast | Estimate | SE | df | *t*-Value | *p*-Value |
|---|---|---|---|---|---|---|---|
| ΔTPH | CINS | No–Yes | 1.62 | 2.28 | 50 | 0.707 | 0.4826 |
|  | EB | No–Yes | 2.46 | 3.47 | 50 | 0.708 | 0.4825 |
|  | FG | No–Yes | 1.95 | 2.14 | 50 | 0.914 | 0.3649 |
|  | HI | No–Yes | −9.48 | 2.62 | 50 | −3.622 | 0.0007 * |
|  | SCI | No–Yes | −1.41 | 2.14 | 50 | −0.662 | 0.5111 |

The diameter distribution of live and dead redbay saplings across all sites reflected the larger trend toward greater densities near suscepts compared to survivors (Figures 7 and 8). In 2008–2009, the drop in live densities going from 4.1–8.0 cm to 8.1–12.0 cm (Figure 7) was less pronounced among dead trees (Figure 8), possibly because the dead trees still reflected the initial diameter distribution and subsequent mortality event in 2007. By 2018–2019, the number of live saplings < 8.0 cm had increased dramatically for both suscepts and survivors (Figure 6), while the number of dead redbay was greatly diminished relative to 2008–2009 and was skewed toward smaller size classes (Figure 7).

Understory Redbay Survey

The clonal midstory resprouts ranged from 0.0 to 197.7 ha$^{-1}$ near survivors, and an identical 0.0 to 197.7 ha$^{-1}$ near suscepts. Both clonal and sexual regeneration tended to be higher near suscepts, but interactions with the site were observed for seedling TPH (Table 3). The suscepts had significantly greater sprouts ($p = 0.0185$) and understory seedlings ($p = 0.00755$) in 2018–2019 (Table 3). The midstory seedlings per hectare showed a site × survivor significant ($p = 0.0184$) effect, primarily driven by two study locations. Pairwise comparisons showed significantly fewer seedlings per hectare near survivors compared to suscepts at FG ($p = 0.0231$) and significantly greater density near survivors compared to suscepts at HI ($p = 0.0178$) (Table 4).

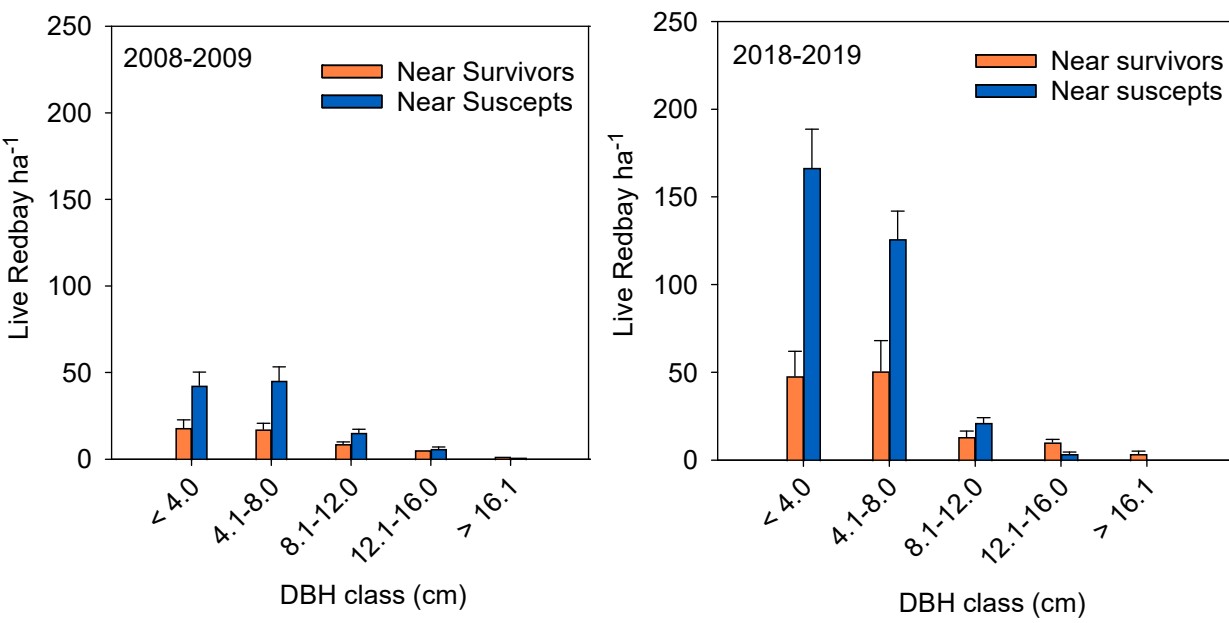

**Figure 7.** Live redbay (ha$^{-1}$) frequency across size classes for plots near survivors and suscepts for all sites. Mean ± SE.

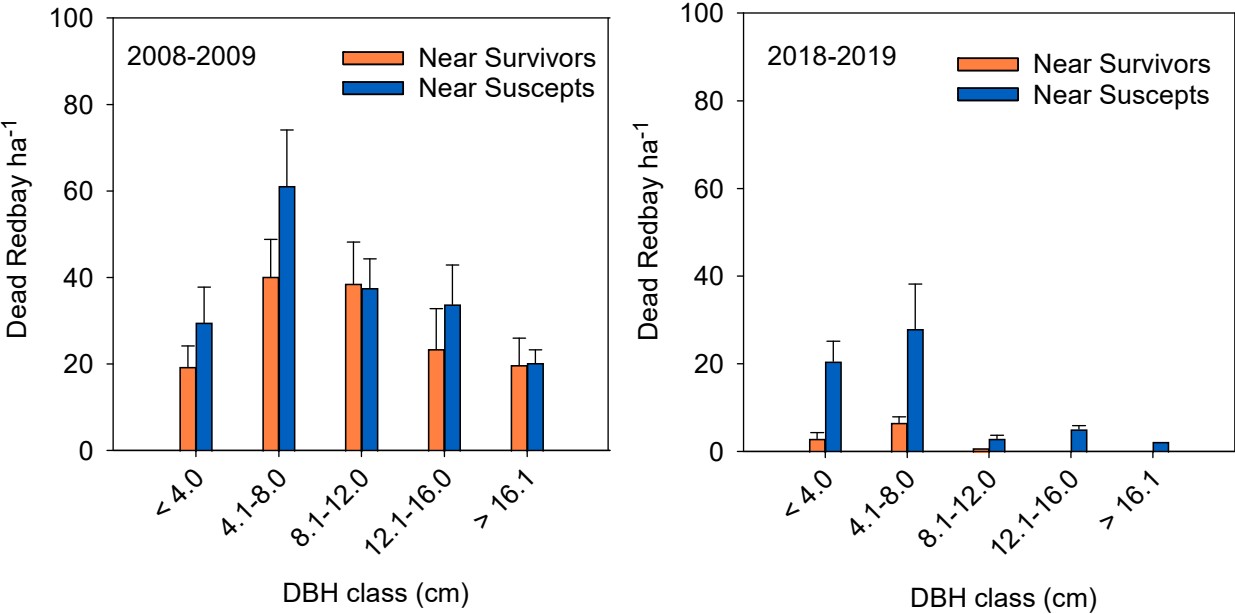

**Figure 8.** Standing dead redbay (ha$^{-1}$) frequency across size classes for plots near survivors and suscepts for all sites. Mean ± SE.

**Table 3.** Contrasts between seedlings and sprouts per hectare (TPH) and understory seedlings (Sdu) for the 2018–2019 sampling period for plots around survivor and suscept trees. Plot values indicate means ± SE, and differences are significant (*) at the level of $p < 0.05$.

|  | Survivor Plots | Suscept Plots | Survivor Effect | Site Effect | Survivor × Site Interaction |
|---|---|---|---|---|---|
| Seedling TPH | 53.4 ± 21.6 | 90.3 ± 14.1 | $p = 0.00264$ * | $p = 8.47 \times 10^{-6}$ * | $p = 0.01844$ * |
| Sprout TPH | 30.3 ± 10.9 | 46.3 ± 7.9 | $p = 0.0185$ * | $p = 1.31 \times 10^{-8}$ * | $p = 0.1118$ |
| Sdu | 4.34 ± 0.99 | 12.00 ± 2.51 | $p = 0.00755$ * | $p = 1.17 \times 10^{-5}$ * | $p = 0.23206$ |

**Table 4.** Pairwise plot comparisons between survivor and suscept plots for seedlings per hectare (TPH) were recorded for the 2018–2019 sampling period. The FC site had no survivors. Values are significant (*) at the level of $p < 0.05$.

|  | Site | Contrast | Estimate | SE | df | *t*-Value | *p*-Value |
|---|---|---|---|---|---|---|---|
| Seedling TPH | CINS | No–Yes | −0.583 | 2.59 | 50 | −0.225 | 0.8230 |
|  | EB | No–Yes | 4.879 | 3.94 | 50 | 1.237 | 0.2217 |
|  | FG | No–Yes | 5.689 | 2.43 | 50 | 2.344 | 0.0231 * |
|  | HI | No–Yes | −7.285 | 2.97 | 50 | −2.451 | 0.0178 * |
|  | SCI | No–Yes | −0.757 | 2.43 | 50 | −0.312 | 0.7563 |

## 4. Discussion

This is the first study to investigate whether redbay survivors are associated with higher rates of tree growth and types of regeneration. Given their continued survival over the years in areas where redbay of their size should theoretically have been killed [11], the survivors presented an opportunity to determine whether genetic tolerance to laurel wilt has allowed them to persist and to observe their associations with redbay regeneration. Overall, the rates of mortality in our study among the original candidates (~3.6%/year) were greater than what has been reported for other deciduous species (<2%/year) in non-disease-affected forest systems [36]. Yet, the presence of a survivor corresponded to elevated growth of nearby saplings, which could be influenced by genetic predisposition or site characteristics, namely sapling density. The continued survival and growth of the saplings near survivors after reaching the susceptible size threshold (Table 1) gives support to the theory that there could be disease tolerance in these individuals as well. Whether this tolerance is inherited or even exists among the population is unknown and should be investigated.

The greater increases in sapling numbers per hectare near suscepts suggest that sapling recruitment may be negatively affected by survivors at these sites. The significantly higher numbers of resprouts per hectare and understory seedlings near suscepts support the idea that regeneration is effectively occurring regardless of the presence of survivors or is increased when the overstory is lost. Notably, flowering and fruiting at young ages has been reported in forest *Persea* spp. [17]. Fruit was observed on an EB redbay measuring 1.52 cm DBH, well below the generally accepted threshold of *X. glabratus* attack [37]. This is the first assessment of redbay regeneration around survivors, but Spiegel and Leege [24] found that at the shrub layer, redbay regeneration was not statistically different between infested and control sites. These findings differ somewhat from those of Cameron et al. [38], who suggested that abundant seedling regeneration was rare where laurel wilt had wiped out redbay stands. Contrary to the results of Evans et al. [25], resprouting does appear to be a capable strategy for reaching sapling sizes. That study, however, focused solely on redbay regeneration in the maritime forests of St. Catherine's Island, which may not be representative of the species throughout its range [38]. No redbay understory was observed at St. Catherine's Island during these surveys, which could reflect a long-term hardwood regeneration failure caused by excessive deer browsing [25].

There were significant interactions between survivors and site in terms of changes in midstory seedlings per hectare. Survivors at HI had higher levels of regeneration compared to suscept plots, suggesting a local effect of survivors on sexual regeneration (Table 2). There could be inter-site, genetic variation in the connection between survival and regeneration. In addition, community assemblages have likely been altered due to the loss of large redbay stems, resulting in subsequent shifts in both understory and midstory composition and cover and natural disturbances, such as hurricanes. Disentangling these possibilities was beyond this study, but we note them as important areas of future research.

In aggregate, the sites appear to be approaching another period of relatively high redbay density, especially near suscepts (Figures 7 and 8). Laurel wilt incidence has been positively correlated with the density of large hosts, especially those larger than 5 and 7.5 cm DBH [39]. Monitoring the status of survivors as nearby saplings grow larger may reveal genotypes with higher degrees of resistance to the pathogen. Increased regeneration may further test existing survivors as disease pressure at these sites still exists [37], meaning that the continued survival of these trees could only be plausible with some sort of mechanisms for evading or withstanding *X. glabratus* attack and *H. lauricola* infection, respectively.

Screening cuttings propagated from the remaining survivors to isolate the most tolerant genotypes has been an ongoing process, and some genotypes have shown consistent survival in artificial inoculations [31]. In particular, the "FGC" genotype has performed very well in artificial inoculation trials [31], while the tree itself is surrounded by numerous saplings in situ [37]. Determining which genotypes show a combination of resistance and regeneration traits may be an effective strategy to isolate the most disease-tolerant germplasm to use for restoration efforts. The long-term restoration feasibility of this germplasm will depend on the durability of tolerance in these genotypes [40]. Major resistance genes in sugar pine (*Pinus lamertiana* Douglas) have been overcome by rapidly evolving strains of *Cronartium ribicola* (J.C. Fisch.) [41]. Fungi can genetically recombine and rapidly adapt to change; however, the lack of genetic diversity in the introduced populations of *X. glabratus* and *H. lauricola* may limit this capability, potentially allowing the tolerance in survivors to be durable [42,43]. Therefore, if disease-resistant redbay is the greatest hope for the species' recovery, then preventing the introduction of a new strain of *H. lauricola* is an imperative step. The monitoring of survivors must continue, as well as the search for other large, healthy redbay, especially in size classes beyond 10.2 cm DBH, as survival past this point has become rare (Figures 7 and 8) to the point that it automatically warrants questions about resistance in the genome. Furthermore, the identification of new genotypes of interest can expand genetic diversity in the current resistance program [33] in conjunction with the ongoing genetic efforts to understand these mechanisms [31,44], determining if genetic linkages exist between survivors and saplings as a function of proximity [45]. Identifying the genes responsible for laurel wilt resistance in redbay and their durability will allow for selective breeding efforts to be undertaken effectively [33,45]. Finally, as with other species that have been affected by invasive species, the outplanting of disease-resistant redbay germplasm should commence on a wider scale, and reforestation potential should be evaluated, specifically to determine if individuals that have performed well in laboratory trials can withstand disease pressure in the field throughout all phases of their life cycle [34,46]. Silvicultural guidelines to ensure seedling survival should be identified. Outplanting trials underway at the Ordway-Swisher Biological Station in Melrose, FL have provided insight into important stock characteristics and soil and moisture regimes conducive to the establishment of disease-resistant redbay [37]. Overall, redbay does not appear to be at risk of extinction given its reproductive resilience, but continued research and effective management will be necessary to facilitate the species' occurrence as a large-diameter tree in the region.

## 5. Conclusions

Despite the large size of survivors, we did not find that it translated to greater recruitment of seedlings. Rather, both seedlings and resprouts were greater in areas with suscepts. Further investigations into genetic linkages between the surviving trees and recruited saplings are needed. A new cohort of redbay appeared to be near the size threshold for *X. glabratus* attack. With continued genomic analyses, disease screenings, and field observations to pinpoint genotypes that may carry genes conferring a high tolerance or resistance to laurel wilt, selective breeding from the new cohort and survivors could help to further support reforestation efforts.

**Author Contributions:** Conceptualization, J.S. and J.V.; methodology, J.S. and J.V.; investigation, M.E.; resources, J.S. and J.V.; writing—review and editing, J.S., J.V. and M.E. All authors have read and agreed to the published version of the manuscript.

**Funding:** This research was funded by the U.S. Department of Agriculture Forest Service (16-DG-11083150-008) "Forest Health Protection—Developing laurel-wilt resistant redbay (*Persea borbonia*) and swamp bay (*Persea palustris*)", the U.S. Department of Agriculture APHIS (AP17PPQFO000C438) FL FY 17—"Mitigating cultural impacts of Laurel Wilt: Year 2-Farm Bill Project 6", and the University of Florida "Jumpstart" award to Vogel and Smith.

**Data Availability Statement:** Data are presented in the paper and exact site location information and demographic data are available upon request.

**Acknowledgments:** We are very grateful to Marc Hughes for his initial research efforts which laid the foundation for this research, as well as the University of Florida Forest Pathology and Forest Ecosystem Science Laboratories for field assistance.

**Conflicts of Interest:** The authors declare no conflicts of interest.

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
