# Peer review of "Redbay (Persea borbonia L. Spreng) Seedling and Sapling Growth and Recruitment Near Trees with and without Putative Resistance to Laurel Wilt Disease"

_forests, doi:10.3390/f15050817_

Round 1
Reviewer 1 Report
Comments and Suggestions for Authors
I have suggested some changes in the attached PDF, which are self-explanatory.
Please consider below points raised in the PDF:
Introduction:
Give the economic/social/atheistic importance.
Only five studies are included in the second last paragraph of the Introduction. Please include a few more, updated ones.
The rationale of the study should be explained.
M & M
Why the original candidate trees were relocated in 2018-19?
Have you studied the insect population at different locations? The insect population might vary at different locations which affects the seedling's survival rate.
The first survival recorded year (2007-09) is not mentioned in M & M.
Check Table 2 and others.
Figure 7 needs to be checked.

Comments on the Quality of English LanguageMinor editing of the English language is required.
Author Response
Please use the attached word file to evaluate our response.

Reviewer 2 Report
Comments and Suggestions for Authors
In term of natural condition, the location of the studied sites should be more specific. Moreover, the term “near” is too general. A more specific distance should be defined in the abstract.
Line 56: it should be “[10-14]”, the same should be applied to other places.
Although the objective is clear, each survey should be mentioned the aim for readers if it is possible
Although the study has been well conducted and its manuscript was also well written, the references are too outdated, i.e. there is only 1 literature in 2023, 2 in 2022, and 1 in 2021. Please try to renew or update the references.
Author Response
Please use the attached file to evaluate our response to your comments.

Reviewer 3 Report
Comments and Suggestions for Authors
The MS “Redbay (Persea borbonia) seedling and sapling demographics 2 near trees with varying levels of resistance to laurel wilt disease” report an important finding and well written. It can published after following minor revision.
Line 2 author name of plant species
Line 9 author name of fungal species
Line 10 author name and order: family of insect follow the same trend throughout the MS
Line 32 please write clear recommendations
Throughout the MS, please write author name of each taxon if appeared first time
In introduction, author should add some earlier report conducted on this topic and how you study is novel in this regard, also add clear objectives of the study
Results should be supported with statistical value.
Please provide clear image of wilt disease/root symptoms that you observed in the field.
In the last of discussion write some paragraph what are the major lever to tackle this disease with supporting references.
Reference style should be as per Journal Guideline.
Comments on the Quality of English LanguageThe MS “Redbay (Persea borbonia) seedling and sapling demographics 2 near trees with varying levels of resistance to laurel wilt disease” report an important finding and well written. It can published after following minor revision.
Line 2 author name of plant species
Line 9 author name of fungal species
Line 10 author name and order: family of insect follow the same trend throughout the MS
Line 32 please write clear recommendations
Throughout the MS, please write author name of each taxon if appeared first time
In introduction, author should add some earlier report conducted on this topic and how you study is novel in this regard, also add clear objectives of the study
Results should be supported with statistical value.
Please provide clear image of wilt disease/root symptoms that you observed in the field.
In the last of discussion write some paragraph what are the major lever to tackle this disease with supporting references.
Reference style should be as per Journal Guideline.
Author Response
Please use the attached file to view our response to your comments.
Thank you,

Reviewer 4 Report
Comments and Suggestions for Authors
Dear Authors,
The present manuscript observed the possible resistance of Redbay to the agent of the laurel wilt disease. The results of the study demonstrated that Redbay regeneration occur regardless of presence of survivors. Unfortunately, the study did not present data that demonstrate correlation or causality in any of observations. Although, the idea of the study is interesting and the results demonstrate value for the field.
Please, see my suggestions below.
Title - Please, verify the TITLE of the manuscript. Besides the misspelling, it is not clear. I would suggest to improve it in a way that could demonstrate the findings of the manuscript.
RESULTS -
Figure 6 and 7 – Something is missing in these figures. What the colors of the legends represent in the figures?
DISCUSSION -
Based in your statement in line 244, what kind of experiment could be done to address this question? Please, a short discussion would be valuable in your discussion.
REFERENCES -
What is the value of “Personal communication”? The manuscript has too many “personal communication”, and all are from the authors of the manuscript. Please, remove them or exchange to a published reference if available.
Author Response
Please use the attached file to see our response to your comments.

Reviewer 5 Report
Comments and Suggestions for Authors
The paper studies Redbay resistance in areas affected by laurel wilt disease. The paper compares population densities and growth under two conditions, presence of resistant and susceptible Redbay individuals. The study presents interesting data to screen for genetic resistance in Redbay but the authors should address some concerns to improve the manuscript:
It is not clear the definition of survivor and susceptible. In line 118, the authors give the following definition: “Candidates that had survived over the last decade are hereafter referred to as “survivors” while those that died will be referred to as “suscepts””. However, even though the authors propose the term susceptible, the trees that did not survive (“suscepts”) might not have died because of laurel wilt disease. In line 119, the authors admit that some tress “decomposed without leaving traces” making it impossible to know is the tree was affected by laurel wilt disease or any other cause. The assumption of any tree “decomposed without leaving traces” as a laurel-wilt-susceptible tree, and any tree surviving as a resistant tree, is problematic.
The second concern is the population density. The authors suggest the “The continued survival and growth of the saplings near survivors after the size of potential infection (Figure 7) indicates that there could be tolerance”. However, Figure 7 does not show “survival and growth of the saplings” but dead Readbay trees per ha. Figure 6 also shows less live Redbay trees per ha near survivors. Since there are “greater densities near suscepts than survivors (line 203)”, these greater densities might explain the increased mortality and not tolerance.
Author Response

(The authors gave the same response as above.)
